# Comprehensive Etiologic Analyses in Pediatric Cochlear Implantees and the Clinical Implications

**DOI:** 10.3390/biomedicines10081846

**Published:** 2022-07-31

**Authors:** Chee-Yee Lee, Pei-Hsuan Lin, Cheng-Yu Tsai, Yu-Ting Chiang, Hong-Ping Chiou, Ko-Yin Chiang, Pei-Lung Chen, Jacob Shu-Jui Hsu, Tien-Chen Liu, Hung-Pin Wu, Chen-Chi Wu, Chuan-Jen Hsu

**Affiliations:** 1Department of Otolaryngology-Head and Neck Surgery, Taichung Tzu Chi Hospital, Buddhist Tzu Chi Medical Foundation, Taichung 42743, Taiwan; allenlee0919@gmail.com; 2School of Medicine, College of Medicine, Tzu Chi University, Hualien 97074, Taiwan; 3Department of Otolaryngology, National Taiwan University Hospital, Taipei 10002, Taiwan; ru3au3@gmail.com (P.-H.L.); ashley.chiang413@gmail.com (Y.-T.C.); risa.chiang@gmail.com (K.-Y.C.); liuent@ntu.edu.tw (T.-C.L.); 4Graduate Institute of Medical Genomics and Proteomics, National Taiwan University College of Medicine, Taipei 10055, Taiwan; leon9139@gmail.com (C.-Y.T.); paylong@ntu.edu.tw (P.-L.C.); jacobhsu@ntu.edu.tw (J.S.-J.H.); 5Hearing, Speech and Cochlear Implant Center, Taichung Tzu Chi Hospital, Buddhist Tzu Chi Medical Foundation, Taichung 42743, Taiwan; redapplepie0307@gmail.com; 6Hearing and Speech Center, National Taiwan University Hospital, Taipei 10002, Taiwan; 7Department of Medical Genetics, National Taiwan University Hospital, Taipei 10002, Taiwan; 8Department of Medical Research, National Taiwan University Hospital Hsin-Chu Branch, Hsin-Chu 30261, Taiwan; 9Department of Otolaryngology, National Taiwan University Hospital Hsin-Chu Branch, Hsin-Chu 30261, Taiwan

**Keywords:** sensorineural hearing impairment, next generation sequencing, cochlear implant, genetic diagnosis

## Abstract

Cochlear implantation is the treatment of choice for children with profound sensorineural hearing impairment (SNHI), yet the outcomes of cochlear implants (CI) vary significantly across individuals. To investigate the CI outcomes in pediatric patients with SNHI due to various etiologies, we prospectively recruited children who underwent CI surgery at two tertiary referral CI centers from 2010 to 2021. All patients underwent comprehensive history taking, next generation sequencing (NGS)-based genetic examinations, and imaging studies. The CI outcomes were evaluated using Categories of Auditory Performance (CAP) and Speech Intelligibility Rating (SIR) scores. Of the 160 pediatric cochlear implantees (76 females and 84 males) included in this study, comprehensive etiological work-up helped achieve clinical diagnoses in 83.1% (133/160) of the patients, with genetic factors being the leading cause (61.3%). Imaging studies identified certain findings in 31 additional patients (19.3%). Four patients (2.5%) were identified with congenital cytomegalovirus infection (cCMV), and 27 patients (16.9%) remained with unknown etiologies. Pathogenic variants in the four predominant non-syndromic SNHI genes (i.e., *SLC26A4, GJB2, MYO15A,* and *OTOF*) were associated with favorable CI outcomes (Chi-square test, *p* = 0.023), whereas cochlear nerve deficiency (CND) on imaging studies was associated with unfavorable CI outcomes (Chi-square test, *p* < 0.001). Our results demonstrated a clear correlation between the etiologies and CI outcomes, underscoring the importance of thorough etiological work-up preoperatively in pediatric CI candidates.

## 1. Introduction

Severe to profound SNHI occurs in approximately 1 in 1000 children [1]. Hearing aids are usually not effective for such severe cases of SNHI. Therefore, surgical intervention with CI is necessary to optimize auditory and speech performance. Bypassing the sensory organ of the inner ear, the CI activates auditory nerve fibers directly, transmits auditory signals through the central neural pathway, and ultimately yields speech understanding in the cortex [2]. Although significant improvements in auditory speech performance [3] and cognitive development [4] can be observed in the majority of CI recipients, there are still about 10% of patients who do not benefit satisfactorily from CIs [5,6].

In its essence, pediatric SNHI is an etiologically heterogeneous condition caused by a plethora of genetic, prenatal, perinatal, and/or environmental factors, and the outcomes in CI are closely related to the underlying etiologies. Infants admitted to the neonatal intensive care unit (NICU) have greater chances of developing SNHI because of the presence of multiple risk factors, such as prematurity and hyperbilirubinemia requiring phototherapy [7]. Specifically, it has been reported that certain functional pathologies (such as auditory neuropathy) [8] and anatomical abnormalities (such as cochlear nerve abnormalities) [9] can compromise the outcomes of CI. In addition, genetic factors, which have been documented as the most common cause of pediatric SNHI [10], are also one of the key determinants of the outcomes of CI [11]. Pathogenic variants in most deafness genes, including *GJB2* [12,13], *SLC26A4* [13,14], *OTOF* [11,15], *MTRNR1* [16], *COCH* [17], and *MYH9* [18], have been associated with favorable outcomes, probably because the pathology is confined to the inner ear. However, in our previous study, we also identified that pathogenic variants in *PJVK* and *PCDH15* are associated with unfavorable outcomes, probably because of the involvement of spiral ganglion neurons (SGNs) and retrocochlear pathologies [19,20].

The ability to accurately predict outcomes is the first step in developing individualized management to enhance CI outcomes in hearing-impaired children. Multidimensional family-centered early intervention in hearing-impaired children is also recommended, as it is proven to be one major driver of CI outcomes [21]. Although research in the past decades has identified several prognostic factors, there is still no reliable prediction tool for clinical use, probably due to the heterogeneous etiologies of pediatric SNHI and the lack of integrated clinical data. To address these complex factors, we performed comprehensive etiological investigations in a large cohort of cochlear implantees, with a view to better delineate the prognostics of CI performances in pediatric SNHI.

## 2. Materials and Methods

### 2.1. Subject Recruitment and Clinical Evaluations

We prospectively recruited children who underwent cochlear implantation at two tertiary referral CI centers in Taiwan (National Taiwan University Hospital and Taichung Tzu Chi Hospital) from 2010 to 2021. All subjects underwent history ascertainment as well as audiological, radiological, and genetic examinations before operation. Virological work-ups for cCMV infection, including blood culture and determination of blood viral load, were performed for patients with clinical symptoms that indicated cCMV infection or positive results from a recently implemented newborn cCMV screening program [22,23]. For pre-operative audiological assessments, tone burst auditory brainstem response, auditory steady-state response, behavioral audiometry, or pure-tone audiometry were performed by experienced audiologists depending on the age or cognitive status of the patient. The cognitive and psycho-behavioral status were evaluated by pediatricians and otologists prior to surgery, and pediatric psychologists were consulted whenever indicated. After surgery, all subjects received regular auditory and speech assessments to determine the outcome after CIs. Patients with an age older than 18 years old, conductive or mixed type hearing impairment, incomplete data collection, or follow-up of less than six months were excluded.

The ethnicity of all subjects was Han Chinese. All subjects and/or their parents provided informed consent before participating in this study. The study protocols were approved by the Research Ethics Committees of both hospitals.

### 2.2. Genetic Examinations

All patients underwent NGS-based examinations that targeted 220 known deafness-related genes using the Illumina Miseq platform (Illumina Inc., San Diego, CA, USA) [24,25,26]. Briefly, the alignment of 2 × 300 bp paired-end reads was performed using BWA-MEM [27] and Picard toolkits version 1.134 (Broad Institute, Cambridge, MA, USA). Variant calling for single nucleotide substitution or small deletions/insertions was implemented using GATK HaplotypeCaller [28] and ANNOVAR [29] software. The highest population allele frequencies (called popmax AF) obtained from the Genome Aggregation Database (ver. 2.1.1, http://gnomad.broadinstitute.org/, last accessed on 3 June 2022) were utilized for variant frequency assessment. Variants categorized as “pathogenic” or “likely pathogenic” based on ACMG guidelines [30] were reported as disease-causing, whereas variants of uncertain significance (VUS) were not regarded as disease-causing. The genetic etiology was confirmed as previously described [25] when the disease-causing variant consisting of the heterozygote was detected in dominant genes; the homozygote or compound heterozygote was detected in recessive genes; hemizygote for male or homozygote for female were detected in X-linked genes; homoplasmy or heteroplasmy were detected in mitochondrial genes. Patients with pathogenic or likely-pathogenic variants in multiple genes where the disease-causing variants could not be determined were excluded.

### 2.3. Imaging Examinations

All patients underwent high-resolution computed tomography (HRCT) of the temporal bone or magnetic resolution imaging (MRI) in the posterior fossa of the brain before cochlear implantation [31]. Abnormalities of the central auditory pathway were investigated with non-contrast brain MRI with a resolution of 0.5 mm thickness, whereas the inner ear structures were investigated with temporal bone HRCT with a resolution of 0.6 mm thickness. The morphology of the cochlea, vestibule, semicircular canal, and vestibular aqueduct, cochlear nerves, as well as cerebral and brainstem abnormalities, were evaluated according to the criteria in the literature [32,33,34]. The images were reviewed by two experienced otologists independently to achieve consensus on the imaging diagnosis.

### 2.4. Assessments of the CI Outcomes

The outcomes in CIs were evaluated using Categories of Auditory Performance (CAP) and Speech Intelligibility Rating (SIR) scores [35,36]. The CAP and SIR scores of the patients were compared with those in the in-house database that documented the auditory and speech development at different postoperative time points in more than 300 patients with CI [37]. Patients with CAP/SIR scores greater than or equal to the median scores at the corresponding postoperative time points in the database were considered to have favorable CI outcomes. For patients using implants for more than five years, which was beyond the scope of our database, CAP > 5 and SIR > 3 were considered favorable outcomes [38]. For those who received bilateral sequential cochlear implantation, data from the first implanted ear were used for outcome analyses. We defined outcome performance groups as unfavorable when both CAP and SIR scores were under the curve, according to the period after cochlear implantation.

### 2.5. Statistical Analysis

The Chi-square test, or Fisher’s exact test, was used to compare the categorical variables, while a *t*-test and an independent-samples Kruskal–Wallis test were used to compare between the continuous and ordinal variables. Univariate logistic regression and multivariate logistic regression were applied to investigate the effects of different variables on the outcomes. All tests were 2-tailed and *p* < 0.05 was considered statistically significant. All statistical analyses were performed using SPSS 25 (IBM SPSS, Inc., Chicago, IL, USA).

## 3. Results

### 3.1. Demographic Data and Etiological Analyses

A total of 160 pediatric cochlear implantees were included in this study (84 males and 76 females). The mean age at implantation was 4.9 years old (median: 3.3 years old, range: 1.0–17.9 years old). There were 96 patients with unilateral CI and 64 patients with bilateral CIs (Table 1).

Of the 160 patients, genetic diagnoses were achieved in 98 patients (61.3%). Causative variants in *SLC26A4* (*n* = 36) were the most prevalent causes of SNHI in our cohort, followed by *GJB2* variants (*n* = 22), *MYO15A* variants (*n* = 10), and *OTOF* variants (*n* = 10). Causative variants in *ILDR1, MTRNR,* and *TECTA*, respectively, were also identified in three other patients with non-syndromic SNHI.

Among the patients of genetic causes, 17 patients (10.6%) exhibited clinical features in addition to SNHI, indicative of syndromic SNHI. The genetic diagnoses in these patients include four with CHARGE syndrome (*CHD7* variants), three with Usher 1B syndrome (*MYO7A* variants), one with suspected Usher 1D syndrome (*CDH23* variants), one with Waardenburg syndrome type 2 (*MITF* variant), one with Waardenburg syndrome type 3 (*PAX3* variant), two with LEOPARD syndrome (*PTPN11* variants), one with suspected Wolfram syndrome (*WFS1* variant)*,* one with dominant optic atrophy (*OPA1* variant)*,* one with Crouzon Syndrome (*FGFR3* variant)*,* one with Stickler syndrome type I (*COL2A1*), and one with DiGeorge syndrome (*TBX1* variant).

Imaging examinations revealed pathological clues in 31 additional patients (19.3%), including 20 patients with cochlear nerve deficiency (CND) and 11 patients with other inner ear malformations (IEMs). Other IEMs included cochlear hypoplasia (*n* = 2), common cavity (*n* = 2), incomplete partition type I (IP-I) (*n* = 2), incomplete partition type II (IP-II) with enlarged vestibular aqueduct (EVA) (*n* = 3), incomplete partition type III (IP-III) (*n* = 1), and isolated EVA (*n* = 1).

Four patients (2.5%) were identified with cCMV infection. All four patients failed newborn hearing screening. Three patients exhibited symptoms of cCMV infection at birth, whereas the other patient only presented SNHI at birth.

In total, our comprehensive genetic, imaging, and viral examinations achieved diagnoses in 133 (83.1%) patients, whereas 27 patients (16.9%) remained with unknown etiologies.

### 3.2. Outcome Analyses

Based on our etiological analyses, we categorized the 160 patients into six groups: non-syndromic SNHI (*n* = 81), syndromic SNHI (*n* = 17), CND (*n* = 20), other IEMs (*n* = 11), cCMV infection (*n* = 4), and unknown etiology (*n* = 27). The mean CAP and SIR scores of the patients in each group are summarized in Table 2.

There were no differences in the age at implantation, gender, laterality, and follow-up period after implantation between the groups, but the auditory and speech performances in terms of CAP and SIR scores differed across groups. Particularly, the mean CAP scores in non-syndromic SNHI (5.9 ± 1.5), other IEMs (5.6 ± 1.2), and unknown (5.9 ± 1.1) groups were better than those in syndromic SNHI (4.4 ± 2.0) and CND (3.1 ± 2.1) groups (independent-samples Kruskal–Wallis test, *p* < 0.001). CND groups also had the worst mean SIR scores (1.7 ± 1.0) when compared to other etiology groups (non-syndromic SNHI: 4.0 ± 1.4, syndromic SNHI: 3.4 ± 1.5, other IEMs: 3.0 ± 1.5, and unknown: 3.8 ± 1.3, independent-samples Kruskal–Wallis test, *p* < 0.001). The mean CAP and SIR scores of the CMV group were 4.5 ± 1.3 and 2.3 ± 1.9, respectively, which were also slightly lower than those of the non-syndromic SNHI group.

We then performed subgroup analyses in patients with pathogenic variants in the four predominant non-syndromic SNHI genes: *SLC26A* (*n* = 36)*, GJB2* (*n* = 22)*, MYO15A* (*n* = 10), and *OTOF* (*n* = 10) (Table 3). The patients of the four groups all exhibited mean CAP > 5 and SIR > 3 without significant difference in CAP and SIR scores between groups. Notably, the mean age of cochlear implantation in patients with *SLC26A4* variants (5.4 ± 3.7 y) was older than those in the other three groups (independent-samples Kruskal–Wallis test, *p* < 0.001), probably reflecting the late-onset progressive or fluctuating nature of SNHI associated with *SLC26A4* variants.

Of the 160 cochlear implantees, 20 and 140 patients were categorized as having unfavorable and favorable outcomes when compared with our in-house database, respectively (Table 4). To clarify the determinant factors of the outcomes, we then compared the demographic features and etiologies between the two groups. Pathogenic variants in the four predominant non-syndromic SNHI genes (i.e., *SLC26A4, GJB2, MYO15A,* and *OTOF*) were associated with favorable CI outcomes (Chi-square test, *p* = 0.023), whereas CND was associated with unfavorable CI outcomes (Chi-square test, *p* < 0.001). The percentages of syndromic SNHI and cCMV infection in the unfavorable outcome group were higher than those in the favorable outcome group (15.0% vs. 10.0% and 5.0% vs. 2.1%, respectively), yet the difference did not reach statistical significance. Three patients with syndromic SNHI, including one with Usher syndrome (*MYO7A* variants), one with LEOPARD syndrome (*PTPN11* variant), and one with DiGeorge syndrome (*TBX1* variant) exhibited unfavorable outcomes with CI.

We then performed logistic regression analyses to investigate the effects of different variables on the CI outcomes (Table 5). In univariate logistic regression, male gender (OR = 0.32, 95% CI: 0.11–0.94, *p* = 0.038) and CND (OR = 0.08, 95% CI: 0.03–0.26, *p* < 0.001) were associated with unfavorable outcomes; whereas pathogenic variants in the four predominant non-syndromic SNHI genes were associated with favorable outcomes (OR = 3.27, 95% CI: 1.13–9.48, *p* = 0.029). In multivariate logistic regression, CND remains the most significant prognostic factor of the CI outcomes (OR = 0.11, 95% CI: 0.03–0.40, *p* < 0.001).

## 4. Discussion

In this study, we demonstrated that comprehensive genetic, imaging, and viral examinations could decipher the clinical diagnoses in 83.1% (133 of 160) of pediatric cochlear implantees. Particularly, genetic factors constituted the most important cause, with 61.3% of cochlear implantees confirmed as having non-syndromic or syndromic SNHI based on our NGS-based genetic examinations. The diagnostic yield of NGS-based genetic examinations significantly surpassed that of conventional genetic examinations, where only common deafness genes were screened. For instance, our previous study that screened four deafness genes could only achieve definite diagnoses in approximately 20% of CI candidates [14]. Seligman et al. screened 17 deafness genes in 100 pediatric CI patients, and identified causative variants in 48 patients (48%). The higher genetic diagnostic rate (61.3%) in this study could be attributed to the scope of genes screened, as our NGS-based examination panel encompassed a total of 220 deafness genes. Given the importance of genetic etiologies in CI patients and the decreasing cost of genomic sequencing, NGS-based genetic examinations may eventually be incorporated into the pre-operative assessment routine for pediatric CI candidates.

Predominant deafness genes in pediatric CI patients differ across populations. The most prevalent deafness genes identified in our cohort include *SLC26A4* (22%), *GJB2* (13%), *MYO15A* (6%), and *OTOF* (6%); whereas those identified in the American patients were *GJB2* (36%)*, SLC26A4* (13%), *MYO7A* (8%), and *MYO15A* (8%) [39]. Nonetheless, as demonstrated in the current and previous studies [14,40], pathogenic variants in all these prevalent deafness genes appeared to be associated with favorable CI outcomes, probably because the pathology is confined to the inner ear and the function of the auditory nerve is spared [14]. These findings have important clinical implications, since the identification of pathogenic variants in these genes represents a good sign for the CI outcomes, which may help accelerate the decision-making process among otologists, audiologists, geneticists, and patients.

In contrast to non-syndromic SNHI, the CI outcomes in patients with syndromic SNHI could be affected by additional handicaps such as vision and cognitive defects. Broomfield et al. investigated the CI outcomes in 38 children with syndromic SNHI, including 10 with Waardenburg syndrome, nine with Usher syndrome, seven with Pendred syndrome, five with Jervell and Lange–Nielsen syndrome, and seven with other syndromes. The outcomes in CI were generally satisfactory, but there was significant variation between and within each syndrome group [41]. Miyagawa et al. investigated the CI outcomes in nine patients with syndromic SNHI, including three with Waardenburg syndrome, two with Usher syndrome, one with Down syndrome, one with Noonan syndrome, one with CHARGE syndrome, and one with Jervell and Lange-Nielsen syndrome. The authors reported that patients with Down syndrome, Noonan syndrome, and Waardenburg syndrome showed comparatively poorer CI outcomes and slower auditory speech development [40]. In this study, we identified 17 CI patients with syndromic SNHI. Unfavorable CI outcomes were observed in three patients with Usher, LEOPARD, and DiGeorge syndromes, respectively. As the CI outcomes of syndromic SNHI patients varied within these small sample size studies, such children should be assessed on an individual basis to ensure a realistic expectation.

The pros and cons of early genetic examination should be fully explained to patients and their parents. For instance, the identification of disease-causing *OTOF* variants can accelerate the clinical decision of cochlear implantation in patients with auditory neuropathy spectrum disorder because favorable CI outcomes can be anticipated and cochlear implantation should be performed whenever indicated without unnecessary delay [42]. Similarly, the identification of disease-causing *SLC26A4* variants can also help tailor an appropriate lifestyle, as hearing deterioration can be prevented by avoiding aggressive exercise and head injury. On the other hand, genetic examinations may also increase stress and anxiety in patients and their parents, particularly in cases of inconclusive or uncertain results.

Among pediatric CI patients with negative or inconclusive genetic results, imaging studies identified pathological clues in 31 patients additionally in this study. On the outcome analyses, CND stood out to be the most significant prognostic factor related to unfavorable auditory speech performances. It has been proposed that the occurrence of CND is related to the origin, migration, nerve growth, neurite pathfinding, and the distribution of neurotrophin during the development of the inner ear [43]. Patients with CND often present with clinically profound SNHI. Not only hearing aids were often ineffective, but the outcomes with CI were also poor in these cases [9,44]. For CND patients who cannot benefit from CI, an auditory brainstem implant is recommended.

In addition to CND, it has been documented that severe IEMs [45], such as cochlear hypoplasia [46] and common cavity [47], might also restrain effective stimulation by the electrodes and curtail the CI function. In this study, we did observe lower CAP and SIR scores in patients with other IEMs compared to those with pure non-syndromic SNHI (Table 2). However, the difference did not reach statistical difference, possibly because of the limited patient number and the minor IEMs (such as IP-II, IP-III or EVA) in most of our patients.

Similarly, we also observed lower CAP and SIR scores in patients with cCMV infection. It has been reported that the CI outcomes in cCMV patients are highly variable and can be affected by the coexisting neurodevelopmental disorders [48,49,50].

The strength of this study lies in that we performed comprehensive genetic, radiological, and virological examinations in a relatively large cohort of pediatric CI patients. We demonstrated that this comprehensive approach could help clarify the etiologies in more than 80% of pediatric CI patients, and that etiologies were closely related to the outcomes.

However, some limitations of this study merit discussion. First, this is an observational study with a single ethnic background. Since the genetic underpinnings of SNHI differ remarkably across populations, prudence should be maintained when the findings of this study are to be extrapolated to other populations. Second, we arbitrarily categorized genetic causes and imaging findings into two different types of etiologies, yet there might be overlap between the two. For instance, EVA or IP-II patients with *SLC26A4* variants were classified into the “non-syndromic SNHI” group, whereas those without *SLC26A4* variants were classified into the “other IEMs” group. For EVA or IP-II patients without *SLC26A4* variants, it has been proposed that there could be hidden *SLC26A4* variants that are beyond the detection capabilities of current sequencing strategies [51], thus these patients might share similar etiologies to those with detectable *SLC26A4* variants. Similarly, patients with CND were arbitrarily categorized into the syndromic SNHI group (e.g., two patients with CHARGE syndrome had CND) and the CND group according to the presence of specific symptoms or signs, yet the CI outcome in the syndromic CND patients might be influenced by the CND to a larger extent than the “syndrome” *per se*. Third, as this study investigated a longitudinal cohort with patients recruited from two different CI centers for more than ten years, certain surgical factors (e.g., the adoption of soft surgery and the preservation of residual hearing) [52,53] and early intervention strategies [21] related to CI outcomes could not be ascertained in every subject and were not included in the analyses.

## 5. Conclusions

With a comprehensive etiological work-up that integrates genetic, imaging, and viral examinations, the clinical diagnoses could be clarified in more than 80% of pediatric CI patients. Genetic causes constitute the leading etiologies, followed by CND and IEMs as evidenced by imaging studies. Genetic causes, particularly pathogenic variants in several highly prevalent deafness genes (e.g., *SLC26A4*, *GJB2*, *MYO15A* and *OTOF*) are associated with favorable CI outcomes, whereas certain imaging findings such as CND are associated with unfavorable outcomes. The correlation between the etiologies and CI outcomes underscores the importance of thorough etiological work-up preoperatively.

## Figures and Tables

**Table 1 biomedicines-10-01846-t001:** Etiologies of the 160 cochlear implantees.

Characteristic	Overall Patients *N* = 160
**Age at implantation, y**	Mean: 4.9, median: 3.3, range: 1.0~17.9
Sex	
Male	84 (52.5%)
Female	76 (47.5%)
**Laterality of implant**	
Unilateral	96 (60.0%)
Bilateral	64 (40.0%)
**Genetic causes**	98 (61.3%)
*SLC26A4*	36
*GJB2*	22
*OTOF*	10
*MYO15A*	10
*ILDR1*	1
*MTRNR1*	1
*TECTA*	1
*CHD7* (CHARGE syndrome)	4
*MYO7A* (Usher IB syndrome)	3
*CDH23* (suspected Usher ID syndrome)	1
*MITF* (Waardenburg syndrome type 2)	1
*PAX3* (Waardenburg syndrome type 3)	1
*PTPN11* (LEOPARD syndrome)	2
*WFS1* (suspected Wolfram syndrome)	1
*OPA1* (Dominant optic atrophy)	1
*FGFR3* (Crouzon syndrome)	1
*COL2A1* (Stickler syndrome)	1
*TBX1* (DiGeorge syndrome)	1
**Imaging diagnoses**	31 (19.4%)
CND	20
Other IEMs	11
Common cavity	2
Cochlear hypoplasia	2
IP-I	2
IP-II with EVA	3
IP-III	1
Isolated EVA	1
**cCMV infection**	4 (2.5%)
**Unknown etiologies**	27 (16.9%)

Abbreviations: cCMV, congenital cytomegalovirus; CND, cochlear nerve deficiency; EVA, enlarged vestibule aqueduct; IEM, inner ear malformation; IP-I, Incomplete partition type I; IP-II, Incomplete partition type II; IP-III, Incomplete partition type III.

**Table 2 biomedicines-10-01846-t002:** Comparison of demographic characteristics and outcome performances according to different etiologies.

	Non-Syndromic SNHI *N* = 81		Syndromic SNHI *N* = 17		CND *N* = 20		Other IEMs *N* = 11		cCMV Infection *N* = 4		Unknown Etiology *N* = 27		
	Mean	SD	Mean	SD	Mean	SD	Mean	SD	Mean	SD	Mean	SD	*p* Value
	Number	%	Number	%	Number	%	Number	%	Number	%	Number	%	
**OP age, y**	4.4	3.8	7.8	5.6	4.3	4.9	4.1	4.4	2.4	0.9	5.9	5.6	0.242
**Gender**													0.176
**Female**	35	43.2	10	58.8	6	30.0	6	54.5	2	50.0	17	63.0	
**Male**	46	56.8	7	41.2	14	70.0	5	45.5	2	50.0	10	37.0	
**Laterality**													0.205
**Unilateral**	47	58.0	11	58.8	15	75.0	6	54.5	2	50.0	15	55.6	
**Bilateral**	34	42.0	6	41.2	5	25.0	5	45.5	2	50.0	12	44.4	
**F/U months**	27.8	22.0	20.5	12.3	19.6	15.0	25.6	21.4	39.5	31.9	32.0	29.9	0.337
**CAP**	5.9	1.5	4.4	2.0	3.1	2.1	5.6	1.2	4.5	1.3	5.9	1.1	<0.001 *
**SIR**	4.0	1.4	3.4	1.4	1.7	1.0	3.0	1.5	2.3	1.9	3.8	1.3	<0.001 ^#^

Chi-square test/Independent-Samples Kruskal–Wallis Test; * Non syndromic = IEM = Unknown> Syndromic > CND; # Non syndromic = Syndromic = IEM = Unknown> CND. The cCMV infection group was excluded for statistical analysis due to limited cases. Abbreviations: CAP, categories of auditory performance; cCMV, congenital cytomegalovirus; CND, cochlear nerve deficiency; F/U, follow up; IEMs, inner ear malformations; SIR, speech intelligibility rating scale; SNHI, sensorineural hearing impairment; y, years old.

**Table 3 biomedicines-10-01846-t003:** Subgroup analyses in patients with pathogenic variants in the four predominant non-syndromic SNHI genes.

	*SLC26A4*		*GJB2*		*MYO15A*		*OTOF*		
	*N* = 36		*N* = 22		*N* = 10		*N* = 10		
	Mean	SD	Mean	SD	Mean	SD	Mean	SD	
	Number	%	Number	%	Number	%	Number	%	*p* Value
**OP age, y**	5.4	3.7	3.8	4.1	2.9	1.1	2.4	1.3	<0.001 *
**Gender**									0.017
**Female**	17	47.2	6	27.3	8	80.0	2	20.0	
**Male**	19	52.8	16	72.7	2	20.0	8	80.0	
**Laterality**									0.070
**Unilateral**	26	72.2	9	40.9	5	50.0	4	40.0	
**Bilateral**	10	27.8	13	59.1	5	50.0	6	60.0	
**F/U months**	31.2	27.4	29.3	17.1	17.9	11.5	24.1	18.9	0.274
**CAP**	6.0	1.5	5.7	1.8	6.1	1.0	6.2	0.8	0.891
**SIR**	4.4	1.3	3.6	1.5	4.2	1.3	3.7	0.9	0.081

* GJB2 = MYO15A = OTOF < SLC26A4; Abbreviations: CAP, Categories of Auditory Performance; F/U, follow up; SIR, Speech Intelligibility Rating; y, years old.

**Table 4 biomedicines-10-01846-t004:** Comparison of demographic features and etiologies between patients with favorable and unfavorable CI outcomes.

Variables	Unfavorable Outcomes *N* = 20		Favorable Outcomes *N* = 140		*p* Value
	Mean/	SD	Mean/	SD	
	Number	%	Number	%	
**OP age, y**	3.4	3.4	5.1	4.7	0.065
**Gender**					0.031
Female	5	25.0	71	50.7	
Male	15	75.0	69	49.3	
**Laterality**					0.196
Unilateral	14	70.0	82	58.6	
Bilateral	6	30.0	58	41.4	
**F/U months**	28.2	14.6	26.5	23.0	0.155
**CAP**	2.9	1.5	5.7	1.6	<0.001
**SIR**	1.5	0.8	3.8	1.4	<0.001
**Four predominant non-syndromic SNHI genes**	5	25	73	52.1	0.023
*SLC26A4*	3	60	33	45.2	
*GJB2*	2	40	20	27.4	
*MYO15A*	0	0	10	13.7	
*OTOF*	0	0	10	13.7	
**Syndromic SNHI**	3	15.0	14	10.0	0.387
**CND**	9	45.0	11	7.9	<0.001
**Other IEMs**	0	0	11	7.9	
**cCMV infection**	1	5.0	3	2.1	0.351
**Unknown etiology**	1	5.0	26	18.6	0.079

Abbreviations: CAP, Categories of Auditory Performance; cCMV, congenital cytomegalovirus; CND, Cochlear nerve deficiency; F/U, follow up; IEMs, inner ear malformations; SIR, Speech Intelligibility Rating; y, years old.

**Table 5 biomedicines-10-01846-t005:** Logistic regression analyses of different variables on the CI outcomes.

	Univariate Analyses		Multivariate Analyses	
Variables	OR (95%CI)	*p* Value	OR (95%CI)	*p* Value
**OP age, y**	1.13 (0.96–1.33)	0.130		
**Gender (male)**	0.32 (0.11–0.94)	0.038	0.30 (0.09–0.95)	0.041
**Laterality (bilateral)**	2.00 (0.69–5.81)	0.203		
**F/U months**	1.00 (0.98–1.02)	0.758		
**Four predominant non-syndromic SNHI genes**	3.27 (1.13–9.48)	0.029	1.78 (0.51–6.27)	0.368
**CND**	0.08 (0.03–0.26)	<0.001	0.11 (0.03–0.40)	<0.001

Abbreviations: CND, cochlear nerve deficiency; F/U, follow up; y, years old.

## Data Availability

The data presented in this study are available on request from the corresponding author. The data are not publicly available due to their containing information that could compromise the privacy of research participants.

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
