# Peer review of "Comprehensive Etiologic Analyses in Pediatric Cochlear Implantees and the Clinical Implications"

_biomedicines, 2022, doi:10.3390/biomedicines10081846_

Round 1
Reviewer 1 Report
This is one of the soo many papers reporting own results on genetic screenings in cochlear implant recipients. I think it is very much out of the scope of this journal.
May I propose a submission to
https://www.mdpi.com/journal/genes/special_issues/Functional_Otogenetics
Please add sensorineural hearing loss to your keywords, its merely the key complaint of the treatment
Author Response
Response to Reviewer 1 Comments:
Thank you so much for your valuable comments on our manuscript (Manuscript ID biomedicines-1778840) entitled “Comprehensive etiologic analyses in pediatric cochlear implantees and the clinical implications.” We have carefully addressed your comments point-by-point as follows.
[Comments of Reviewer 1]
Point 1: May I propose a submission to
https://www.mdpi.com/journal/genes/special_issues/Functional_Otogenetics
Response 1: Thank you very much for your valuable comment. By demonstrating the applications of genetic examinations in the clinical decision making for cochlear implantation, we think our findings in this article highlighted “the benefit from the vast amounts of information that can be garnered from genetic work in this field” and fulfilled the scope #2 of this special issue (https://www.mdpi.com/journal/biomedicines/special_issues/Genetic_hearing). We hope that this revised version can be accepted for publication in this special issue of the Journal of Biomedicines.
Point 2: Please add sensorineural hearing loss to your keywords, its merely the key complaint of the treatment.
Response 2: Thank you very much for your precious comment. According to the reviewer’s suggestion, we have added the term “sensorineural hearing loss” to our keywords in the manuscript.
(Keywords; Page 1, line 45)
Reviewer 2 Report
The authors wrote an article about the genomic analyses in pediatric cochlear implantees and the clinical implications. The article is very interesting, well written and the topic is hot.
I'm an ENT surgeon, I can value only the clinical part, but I can not judge the genetic part. I have some suggestion that can improve the quality of the manuscript and give to it a better resonance in the literature.
1. In the introduction, please talk more about the possibility to find children with hearing loss in NICU. Use this reference: Gazia F, Abita P, Alberti G, Loteta S, Longo P, Caminiti F et al. NICU Infants & SNHL: Experience of a western Sicily tertiary care centre. Acta Medica Mediterranea. 2019, 35(2): 1001-7
2. In the part of methods, please talk more about the kind of surgery performed to patients. They have a CI soft surgery with a round window approach ? This is very important because soft surgery can save residual hearing and the children could have a better quality of hearing. Use this reference: Freni F, Gazia F, Slavutsky V, Scherdel EP, Nicenboim L, Posada R et al. Cochlear Implant Surgery: Endomeatal Approach versus Posterior Tympanotomy. Int J Environ Res Public Health. 2020 Jun 12;17(12):E4187.
3. In the statistical part, please use the Kolmorog-Smirnov test to value the normality of values before use T-test
1.
Author Response
Response to Reviewer 2 Comments:
Thank you so much for your valuable comments on our manuscript (Manuscript ID biomedicines-1778840) entitled “Comprehensive etiologic analyses in pediatric cochlear implantees and the clinical implications.” We have carefully addressed your comments point-by-point as follows.
[Comments of Reviewer 2]
Point 1: In the introduction, please talk more about the possibility to find children with hearing loss in NICU. Use this reference: Gazia F, Abita P, Alberti G, Loteta S, Longo P, Caminiti F et al. NICU Infants & SNHL: Experience of a western Sicily tertiary care centre. Acta Medica Mediterranea. 2019, 35(2): 1001-7
Response 1: Thank you very much for the precious comment. We have revised the manuscript following the reviewer’s comment in the Introduction section.
(Introduction; Page 2, lines 58-62)
- In its essence, pediatric SNHI is an etiologically heterogeneous condition caused by a plethora of genetic, prenatal, perinatal, and/or environmental factors, and the outcomes in CI are closely related to the underlying etiologies. Infants admitted to the neonatal intensive care unit have greater chances of developing SNHI, because of the presence of multiple risk factors, such as prematurity and hyperbilirubinemia requiring phototherapy [7].
Point 2: In the part of methods, please talk more about the kind of surgery performed to patients. They have a CI soft surgery with a round window approach ? This is very important because soft surgery can save residual hearing and the children could have a better quality of hearing. Use this reference: Freni F, Gazia F, Slavutsky V, Scherdel EP, Nicenboim L, Posada R et al. Cochlear Implant Surgery: Endomeatal Approach versus Posterior Tympanotomy. Int J Environ Res Public Health. 2020 Jun 12;17(12):E4187.
Response 2: Thank you for this important question. As the reviewer pointed out correctly, soft surgery can save residual hearing and improve the quality of hearing. In our previous study, we also found that the size of round window opening and the speed of electrode insertion were significantly correlated with the preservation of residual hearing (PMID: 32176129). However, as the subjects of the current study were recruited from two different CI centers over > 10 years, we could not ascertain the adoption of soft surgery in every subject. We agree with the reviewer that this is a limitation, and we appreciate the opportunity to clarify this point in the “Discussion” section as follows.
(Discussion; Pages 9-10, lines 351-355)
→ Third, as this study investigated a longitudinal cohort with patients recruited from two different CI centers for more than ten years, certain surgical factors (e.g., the adoption of soft surgery and the preservation of residual hearing) [54, 55] and early intervention strategies [22] related to CI outcomes could not be ascertained in every subject and were not included in the analyses.
Point 3: In the statistical part, please use the Kolmorog-Smirnov test to value the normality of values before use T-test.
Response 3: Thank you for your precious comment. We applied the Kolmorog-Smirnov test to verify the normality of the continuous variables. We found that some of variables were not in normal distribution, so we used the non-parametric Mann-Whitney U test for comparison in Table 4 instead. The p-value remains no statistically significant difference in the OP age and F/U months; while both CAP and SIR remain with p-value <.001 in Table 4. We appreciate the opportunity to revise Table 4 as follows.
(Table 4; Page 7, line 243)
→ Table 4. Comparison of demographic features and etiologies between patients with favorable and unfavorable CI outcomes.
|
Variables
|
Unfavorable outcomes N=20 |
Favorable outcomes N=140 |
P value |
|||
|
|
Mean/ |
SD |
Mean/ |
SD |
|
|
|
|
Number |
% |
Number |
% |
|
|
|
OP age, y |
3.4 |
3.4 |
5.1 |
4.7 |
0.065 |
|
|
F/U months |
28.2 |
14.6 |
26.5 |
23.0 |
0.155 |
|
|
CAP |
2.9 |
1.5 |
5.7 |
1.6 |
<.001 |
|
|
SIR |
1.5 |
0.8 |
3.8 |
1.4 |
<.001 |
|
Reviewer 3 Report
This article describes the results of genetic testing, imagining and virological examination of a large group of cochlear implant recipients.
The abstract represents the article. The article contains detailed data analysis. In discussion part authors compare the results with other published results in the field, highlight strength and limitations of the study.
The article might be recommended for publication
Author Response
Response to Reviewer 3 Comments:
Thank you so much for your valuable comments on our manuscript (Manuscript ID biomedicines-1778840) entitled “Comprehensive etiologic analyses in pediatric cochlear implantees and the clinical implications.” We appreciate it very much.
[Comments of Reviewer 3]
Point 1: This article describes the results of genetic testing, imagining and virological examination of a large group of cochlear implant recipients. The abstract represents the article. The article contains detailed data analysis. In discussion part authors compare the results with other published results in the field, highlight strength and limitations of the study. The article might be recommended for publication.
Response 1: Thank you very much for your kind comments. We really appreciate it. We hope that this revised version can be accepted for publication in the Journal of Biomedicines.
Reviewer 4 Report
In their manuscript "comprehensive genomic analyses in pediatric cochlear implantees and the clinical implcations" Lee et al. present non epidemiological, however, prospectively collected data on 160 pediatric cochlear implantees including stringent and clearly defined CI outcome parameters. The study incoprorates data on patient history , next generation sequencing results and results of cCT and cMRI scans. Overall the manuscript is written well, the introduction section captures nearly all important topics, methods and results are presented adequate and the discussion section is clearly supported by the findings. The presented data and their interpretation are highly relevant and fit well to the chosen Journal. Overall the reviewer clearly recommends acceptance after adresing several minor but however important points:
1) As the study adds more than just "comprehensive genomic analyses" the reviewer recommends to think about a new heading better describing the overall important findings.
2) Page 2, lines 58-59: "... certain physiological pathologies ..." maybe functional instead of physiological?
3) Introduction overall/Limitations: One major limitation of the study are the missing data on Early Intervention strategies. As family centered early intervention is prooven to be one major driver of CI outcome (for example: Holzinger et al. J Clin Med 2022, 11(6), 1548) this fact should be at least mentioned in the introduction section and should be added to the limitations. However, if data on early intervention are availbale they would clearly improve the overall impact of the presented data and should be added.
4) Materials/methods: Page 2 line 87: please descrbe how the cognitive status of the patients was evaluated.
5) Materials/methods: Page 3 lines 99-109: please describe shortly how you managed variants of unclear significance, possible multiple pathogenetic variants.
6) Table 1: please add the age range and median
7) Results: Page 4, line 155: the reviewer would recommend to introduce the category "non-syndromic mimics" (e.g. Usher syndrome yet without retinitis, etc.)
8) Results Page 5, line 159: regarding the WFS1 variant - does the affected patient really show the clinical wolfram syndrome criteria?
9) Results Page 5, lines 176-179: the reviewer recommends to split the group "syndromic" into "syndromic" and "non-syndromic mimics"; in addition to split the group CND in "CND plus [syndromic CND]" and CND only [non-syndromic].
10) Discussion, Page 8. The reviewer misses a short paragraph on pros and cons of early genetic testing for HL in general including parents perspectives.
Author Response
Response to Reviewer 4 Comments:
Thank you so much for your valuable comments on our manuscript (Manuscript ID biomedicines-1778840) entitled “Comprehensive etiologic analyses in pediatric cochlear implantees and the clinical implications.” We have carefully addressed your comments point-by-point as follows.
[Comments of Reviewer 4]
Point 1: As the study adds more than just "comprehensive genomic analyses" the reviewer recommends to think about a new heading better describing the overall important findings.
Response 1: Thank you very much for the precious comment. We have revised the title of manuscript from "comprehensive genomic analyses" to "comprehensive etiologic analyses" according to the reviewer’s comment.
(Title; Page 1, line 2)
→ Comprehensive etiologic analyses in pediatric cochlear implantees and the clinical implications.
Point 2: Page 2, lines 58-59: "... certain physiological pathologies ..." maybe functional instead of physiological?
Response 2: Thank you very much for the suggestion. We have corrected the word following the reviewer’s comment.
(Introduction; Page 2, line 62)
→ Specifically, it has been reported that certain functional pathologies (such as auditory neuropathy) [7] and anatomical abnormalities (such as cochlear nerve abnormalities) [8] can compromise the outcomes in CI.
Point 3: Introduction overall/Limitations: One major limitation of the study are the missing data on Early Intervention strategies. As family centered early intervention is proven to be one major driver of CI outcome (for example: Holzinger et al. J Clin Med 2022, 11(6), 1548) this fact should be at least mentioned in the introduction section and should be added to the limitations. However, if data on early intervention are available they would clearly improve the overall impact of the presented data and should be added.
Response 3: Thank you very much for the precious comment. We agree with the reviewer that a major limitation of the study are the missing data on early intervention strategies. We appreciate the opportunity to clarify this point in both the Introduction and Discussion sections.
(Introduction; Page 2, lines 74-76)
→ Multidimensional family centered early intervention in hearing-impaired children is also recommended, as it is proven to be one major driver of CI outcomes [22].
(Discussion; Pages 9-10, lines 351-355)
→ Third, as this study investigated a longitudinal cohort with patients recruited from two different CI centers for more than ten years, certain surgical factors (e.g., the adoption of soft surgery and the preservation of residual hearing) [54, 55] and early intervention strategies [22] related to CI outcomes could not be ascertained in every subject and were not included in the analyses.
Point 4: Materials/methods: Page 2 line 87: please describe how the cognitive status of the patients was evaluated.
Response 4: Thank you very much for important question. The cognitive and psycho-behavioral status of the patients were evaluated by pediatricians and otologists prior to surgery, and pediatric psychologists were consulted whenever indicated. We appreciate the opportunity to clarify this point as follows.
(Materials and Methods; Page 2, lines 94-96)
→ The cognitive and psycho-behavioral status were evaluated by pediatricians and otologists prior to surgery, and pediatric psychologists were consulted whenever indicated.
Point 5: Materials/methods: Page 3 lines 99-109: please describe shortly how you managed variants of unclear significance, possible multiple pathogenetic variants.
Response 5: Thank you very much for the crucial questions. Variants of uncertain significance (VUS) were not regarded as disease-causing in this study. Patients with pathogenic or likely-pathogenic variants in multiple genes where the disease-causing variants could not be determined were excluded from further analyses. We appreciate the opportunity to clarify these points as follows.
(Materials and Methods; Page 3, lines 112-115 & 119-121)
→ Variants categorized as “pathogenic” or “likely pathogenic” based on ACMG guideline [31] were reported as disease-causing, whereas variants of uncertain significance (VUS) were not regarded as disease-causing.
→ Patients with pathogenic or likely-pathogenic variants in multiple genes where the disease-causing variants could not be determined were excluded.
Point 6: Table 1: please add the age range and median
Response 6: Thank you very much for the comment. We have added the information of the age range and median age in the Results section and Table 1.
(Results; Page 4, lines 157-158 and Table 1; Page 4, line 160)
→ The mean age at implantation was 4.9 years old (median: 3.3 years old, range: 1.0-17.9 years old).
Table 1. Etiologies of the 160 cochlear implantees.
|
Characteristic
|
Overall patients N=160 |
|
Age at implantation, y |
Mean: 4.9, median: 3.3, range: 1.0~17.9 |
Point 7: Page 4, line 155: the reviewer would recommend to introduce the category "non-syndromic mimics" (e.g. Usher syndrome yet without retinitis, etc.)
Response 7: As the reviewer pointed out correctly, patients with certain types of syndromic SNHI might present hearing loss as the only clinical feature initially, mimicking non-syndromic SNHI (i.e, "non-syndromic mimics", such as Usher syndrome yet without retinitis, etc.). Following the reviewer’s suggestions, we rechecked the clinical data of our syndromic SNHI patients carefully. The 3 patients with MYO7A variants were from a multiplex family, and all of them developed hearing loss and retinitis pigmentosa and were confirmed with Usher IB syndrome. The patient with CDH23 variants developed severe photophobia in addition to hearing loss, and Usher ID syndrome was suspected by the ophthalmologist. The patient with WFS1 variant also exhibited some signs of optic nerve atrophy on ophthalmic examinations; however, she did not develop diabetes insipidus or diabetes mellitus. In brief, all of our syndromic SNHI patients demonstrated some signs in addition to pure hearing loss, although their clinical features might not completely fulfill the diagnostic criteria of certain types of syndromic SNHI. Following the guidance of the reviewer’s suggestion, we rephrase our description on two patients: “suspected Usher 1D syndrome” for the patient with CDH23 variants and “suspected Wolfram syndrome” for the patient with WFS1 variant. We also appreciate the chance to add some additional information of these patients in the Results section, Table 1 & Table S1.
(Results; Page 5, line 174, and Page 4, line 160 (Table 1), and Patents; Page 10, lines 366-367, Supplementary Materials, The following supporting information can be downloaded at: www.mdpi.com/xxx/s1.)
→ …, 1 with suspected Usher 1D syndrome (CDH23 variant), …
Table 1. Etiologies of the 160 cochlear implantees.
|
Characteristic
|
Overall patients N=160 |
|
Genetic causes SLC26A4 GJB2 OTOF MYO15A ILDR1 MTRNR1 TECTA CHD7 (CHARGE syndrome) MYO7A (Usher IB syndrome) CDH23 ( suspected Usher ID syndrome) MITF (Waardenburg syndrome type 2) PAX3 (Waardenburg syndrome type 3) PTPN11 (LEOPARD syndrome) WFS1 (suspected Wolfram syndrome) OPA1 (Dominant optic atrophy) FGFR3 (Crouzon syndrome) COL2A1 (Stickler syndrome) TBX1 (DiGeorge syndrome) |
98 (61.3%) 36 22 10 10 1 1 1 4 3 1 1 1 2 1 1 1 1 1 |
Table S1. Detailed information of the 98 patients of genetic causes.
|
NO. |
Gender |
Genotype/ phenotype |
CT/MRI findings |
OP age, y |
Laterality |
CAP |
SIR |
F/U months |
|
DE6263 |
M |
MYO7A/ Usher, multiplex family |
normal CT and MRI |
1.4 |
B |
6 |
3 |
33.0 |
|
DE6270 |
M |
MYO7A/ Usher, multiplex family |
normal CT and MRI |
16.1 |
L |
3 |
2 |
25.0 |
|
DE6264 |
F |
MYO7A/ Usher, multiplex family |
normal CT and MRI |
14.3 |
L |
2 |
2 |
9.0 |
|
DE6560 |
F |
CDH23/ Suspected Usher, severe photophobia with vision problem |
normal CT and MRI |
5.1 |
R |
7 |
5 |
24.0 |
Point 8: Results Page 5, line 159: regarding the WFS1 variant - does the affected patient really show the clinical wolfram syndrome criteria?
Response 8: The reviewer is very insightful in pointing out that patients with pathogenic WFS1 variants do not necessarily develop full-blown Wolfram syndrome. In addition to hearing loss, our patient also exhibits some signs of optic nerve atrophy on ophthalmic examinations; however, she does not develop diabetes insipidus or diabetes mellitus yet. We think that although this patient does not fulfill the diagnostic criteria of Wolfram syndrome completely, she is indeed a patient with syndromic hearing loss. We appreciate the chance to rephrase our description as “suspected Wolfram syndrome” in the Results section, Table 1, and Table S1.
(Page 5, line 176, and Results; Page 4, line 160 (Table 1), and Patents; ; Page 10, lines 366-367, Supplementary Materials, The following supporting information can be downloaded at: www.mdpi.com/xxx/s1. )
→ …, 1 with suspected Wolfram syndrome (WFS1 variant), …
Table 1. Etiologies of the 160 cochlear implantees.
|
Characteristic
|
Overall patients N=160 |
|
Genetic causes SLC26A4 GJB2 OTOF MYO15A ILDR1 MTRNR1 TECTA CHD7 (CHARGE syndrome) MYO7A (Usher IB syndrome) CDH23 ( suspected Usher ID syndrome) MITF (Waardenburg syndrome type 2) PAX3 (Waardenburg syndrome type 3) PTPN11 (LEOPARD syndrome) WFS1 (suspected Wolfram syndrome) OPA1 (Dominant optic atrophy) FGFR3 (Crouzon syndrome) COL2A1 (Stickler syndrome) TBX1 (DiGeorge syndrome) |
98 (61.3%) 36 22 10 10 1 1 1 4 3 1 1 1 2 1 1 1 1 1 |
Table S1. Detailed information of the 98 patients of genetic causes.
|
NO. |
Gender |
Genotype/ phenotype |
CT/MRI findings |
OP age, y |
Laterality |
CAP |
SIR |
F/U months |
|
DE5729 |
F |
WFS1/ suspected Wolfram syndrome with optic nerve atrophy and deafness |
normal CT and MRI |
2.6 |
R |
6 |
4 |
10.0 |
Point 9: Results Page 5, lines 176-179: the reviewer recommends to split the group "syndromic" into "syndromic" and "non-syndromic mimics"; in addition to split the group CND in "CND plus [syndromic CND]" and CND only [non-syndromic].
Response 9: Thank you very much for the important comments.
- As for splitting the group “syndromic” into “syndromic” and “non-syndromic mimics”, we don’t have patients classified as “non-syndromic mimics” in our cohort. As responded to Point 7, although their clinical features might not completely fulfill the diagnostic criteria of certain types of syndromic SNHI, all of our syndromic SNHI patients demonstrated some signs in addition to pure hearing loss.
- Meanwhile, patients with syndromic CND were classified into the syndromic group (e.g., two patients with CHARGE syndrome had CND) in this study, whereas “CND” in this study refers to “non-syndromic CND” merely. We agree with the reviewer that it might be interesting to compare the outcomes between "CND plus [syndromic CND]" and “CND only [non-syndromic]”. However, it is rather difficult to perform further statistical analyses on the split groups with very limited patient numbers. This is indeed a limitation of this study, and we appreciate the chance to address this point as follows.
(Discussion; Page 9, lines 347-351)
→ Second, we arbitrarily categorized genetic causes and imaging findings into two different types of etiologies, yet there might be overlapping between the two. For instance, EVA or IP-II patients with SLC26A4 variants were classified into the "non-syndromic SNHI” group, whereas those without SLC26A4 variants were classified into the "other IEMs” group. …. Similarly, patients with CND were arbitrarily categorized into the syndromic SNHI group (e.g., two patients with CHARGE syndrome had CND) and the CND group according to the presence of specific symptoms or signs, yet the CI outcome in the syndromic CND patients might be influenced by the CND to a larger extent then the “syndrome” per se.
Point 10: Discussion, Page 8. The reviewer misses a short paragraph on pros and cons of early genetic testing for HL in general including parents perspectives.
Response 10: Thank you very much for the precious comment. We have added a short paragraph in the Discussion section to address this point.
(Discussion; Page 9, lines 302-311)
→ The pros and cons of early genetic examination shall be fully informed to patients and their parents. For instance, the identification of disease-causing OTOF variants can accelerate the clinical decision of cochlear implantation in patients with auditory neuropathy spectrum disorder, because favorable CI outcomes can be anticipated and cochlear implantation should be performed whenever indicated without unnecessary delay [44]. Similarly, the identification of disease-causing SLC26A4 variants can also help tailor appropriate lifestyle, as hearing deterioration can be prevented by avoiding aggressive exercise and head injury. On the other hand, genetic examinations may also increase the stress and anxiety in patients and their parents, particularly in cases of inconclusive or uncertain results.